# Recent Progress of Residual Stress Distribution and Structural Evolution in Materials and Components by Neutron Diffraction Measurement at RSND

**Fangjie Mo [1,2], Guangai Sun [1,*], Jian Li [1], Changsheng Zhang [1], Hong Wang [1], Ying Chen [3], Zhao Liu [1], Zukun Yang [1], Hongjia Li [1], Zhaolong Yang [1], Beibei Pang [1], Yalin Huang [1], Yi Tian [1], Jian Gong [1], Bo Chen [1] and Shuming Peng [1,2,*]**

[1] Key Laboratory for Neutron Physics of Chinese Academy of Engineering Physics, Institute of Nuclear Physics and Chemistry, Mianyang 621999, China; mofangjie17@gscaep.ac.cn (F.M.); lijian1980@caep.cn (J.L.); john_yzm@163.com (C.Z.); wanghong87730@163.com (H.W.); liuzhao7322@163.com (Z.L.); zxyan815@163.com (Z.Y.); lihongjia_caep@126.com (H.L.); yangzhl07@163.com (Z.Y.); sifan12459@163.com (B.P.); huangyalin10@163.com (Y.H.); tyoreo@163.com (Y.T.); gongjian@hotmail.com (J.G.); chenbo_58@163.com (B.C.)

[2] Shanghai EBIT Lab, Key Laboratory of Nuclear Physics and Ion-beam Application, Institute of Modern Physics, Department of Nuclear Science and Technology, Fudan University, Shanghai 200433, China

[3] Department of Materials Science, Fudan University, Shanghai 200433, China; 14307130351@fudan.edu.cn

\* Correspondence: guangaisun_80@163.com (G.S.); pengshuming@caep.cn (S.P.); Tel.: +86-08162493337 (G.S.); +86-08162493835 (S.P.)

**Abstract:** Neutron diffraction is an effective and nondestructive method to investigate inner structure and stress distribution inside bulk materials and components. Compared with X-ray diffraction, neutron diffraction allows a relatively high penetration depth and covers a larger gauge volume, enabling it to measure the lattice structure and three-dimensional (3D) distribution of residual stress deep inside thick sample materials. This paper presents the recent development of a Residual Stress Neutron Diffractometer (RSND) at the Key Laboratory for Neutron Physics of the Chinese Academy of Engineering Physics, Institute of Nuclear Physics and Chemistry, Mianyang, China. By integrating multiple instruments such as a loading frame, Kappa goniometer, and coupling system, the RSND was constructed as a suitable platform for various neutron diffraction experiments, including residual stress measurement, in situ observation, and texture analysis. Neutron diffraction measurement can be used to study various materials such as steels, aluminum alloys, and titanium alloys, as well as various components such as turbine discs and welding parts. An evaluation method for both polycrystalline and monocrystalline materials was developed, and this method was found to have the capability of solving an agelong technical challenge in characterizing monocrystalline materials. Furthermore, by introducing a texture and thermomechanical coupling system, it is now possible to make effective in situ observations of the structural evolution in single crystal materials under high-temperature tensile conditions.

**Keywords:** residual stress; structural evolution; single crystal; components; neutron diffraction; in situ measurement

## 1. Introduction

Residual stress, which originates from the inconsistent distortion between different regions, can be defined as self-balanced internal stress remaining inside materials and components after external stress or non-uniform temperature fields are eliminated. Processes such as machining, heat

treatment, deformation, and phase transition may cause such tress [1,2]. Residual stress can be divided into three categories according to their characteristic self-equilibrate scale, as is shown in Figure 1. Type I stress is known as macrostress, which varies continuously upon macroscopic components. Interangular stress (type II) is of grain scale, while type III stress is of atomic scale. Residual stress in general is a crucial parameter in the field of engineering as it helps estimate properties such as mechanical properties, thermal behavior, service life, etc. [3]. Its precise evaluation is the basis for further performance evaluation. For example, the rate of plastic deformation is closely related to the intensity and distribution of residual stress inside components [4]. Therefore, it is meaningful to accurately determine the distribution and evolution of residual stress when engineering materials and components are subjected to specific conditions.

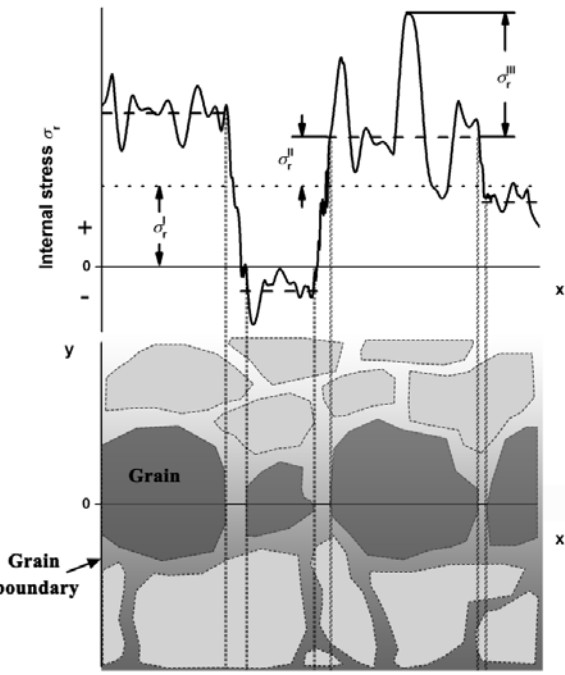

**Figure 1.** The schematic diagram for different types of residual stress where $\sigma_r^I$, $\sigma_r^{II}$, and $\sigma_r^{III}$ respectively stand for type I, type II, and type III stresses.

Neutron diffraction is an effective and nondestructive method to measure residual stress, based on the following principles. The existence of residual stress inside materials will lead to an increase or decrease of the $d$-spacing of lattice planes due to vertical tension or compression, respectively. Together with $d_0$, which is the initial spacing under a stress-free state, the strain $\varepsilon$ of the plane can be obtained according to the equation: $\varepsilon = \frac{d-d_0}{d_0}$. The diffraction method determines the $d$-spacing of lattice planes based on Bragg's law: $n\lambda = 2d \sin\theta$, where $\lambda$ is the incident neutron wavelength and $\theta$ is the diffraction angle of the measured plane. Lattice strain $\varepsilon$ can be calculated by the shift of the diffraction angle $\Delta\theta$, as shown in the following equation:

$$\varepsilon = \frac{d - d_0}{d_0} = -\Delta\theta \cot\theta_0, \tag{1}$$

where $\theta_0$ is the diffraction angle of a sample under its initial state. This equation applies only to the evaluation of macrostresses (type I) and interangular stresses (type II) inside polycrystalline materials, while the measurement of single crystal materials is based on a more complex equation:

$$\begin{aligned}
\varepsilon = \frac{d-d_0}{d_0} = &\; \varepsilon_{11} \cos^2\varphi \cos^2\chi + \varepsilon_{22} \sin^2\varphi \cos^2\chi + \varepsilon_{33} \sin^2\chi \\
&+ \varepsilon_{12} \sin 2\varphi \cos^2\chi + \varepsilon_{13} \cos\varphi \sin 2\chi + \varepsilon_{23} \sin\varphi \sin 2\chi,
\end{aligned} \tag{2}$$

where $\varphi$ and $\chi$ are the rotational and tilting angles of the sample, respectively (Figure 2), and $\varepsilon_{ij}$ is the component of the strain tensor in the sample coordinate system [3]. At least six sets of neutron diffraction data are needed to determine the residual stress inside a single crystal sample.

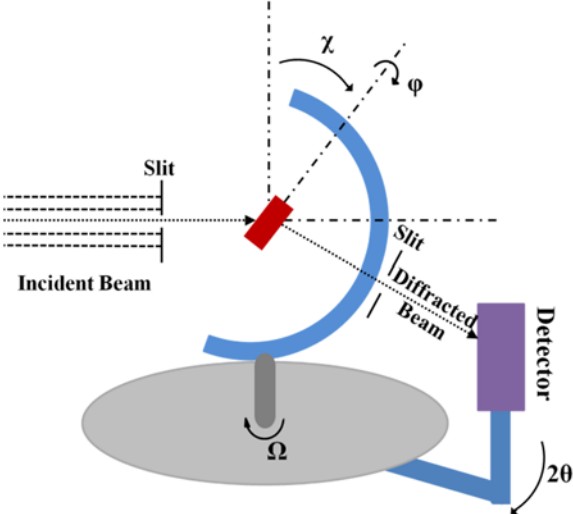

**Figure 2.** Schematic diagram of the geometry of the setup, where $\varphi$ and $\chi$ represents respectively the sample's rotation and tilting, whilst $\Omega$ represents the sample table's rotation.

Compared with other nondestructive methods, such as X-ray diffraction for example, neutron diffraction exhibits many advantages [5]. Firstly, neutron diffraction allows a relatively high penetration depth, which can reach an average depth of several centimeters in common engineering materials, so that the depth is relatively deeper than that of X-ray methods ($\mu$m) [6,7]. Neutron diffraction can thus be used to analyze the inner structures of even heavy element materials. Secondly, the wavelength of neutrons is adjustable through the control of the take-off angle and monochromator, whereas the wavelength of X-rays can only be selected by changing the X-ray targets, such as Cu (1.54 Å), Cr (2.29 Å), Co (1.79 Å), and Mo (0.71 Å). The intensity of neutron beams is also controllable; a high-intensity neutron beam is therefore able to enhance the efficiency of in situ experiments. Moreover, neutron diffraction reflects the structure of a larger volume than X-ray diffraction, not only reducing errors caused by singular spots, but also making it an effective method to study three-dimensional (3D) residual stress inside materials and components [8].

Worldwide, there are many neutron scattering platforms. Akita et al. used the in situ neutron diffraction equipment on KOWARI strain scanner at the Australia Nuclear Science Technology Organization (ANSTO, Sydney, Australia) to measure the residual stress of a dissimilar weld joint. The effect of thermal residual stress on yield strength was clearly discussed [9]. Haque et al. measured the residual stress in riveted joints of high-strength steel using neutron diffraction on KOWARI. The results showed that the residual stress at the center of the bore was compressive, while that away from the center became tensile. The compressive stress was greater for a thin joint in a rivet leg than that for a thick joint [10]. Brown et al. researched the residual stress, dislocation density, and texture of U-10Mo alloy on the Spectrometer for Materials Research at Temperature and Stress (SMARTS) at the Los Alamos Neutron Science Center (LANSCE, Los Alamos, NM, USA). They found that the thermal expansion mismatch was the main reason affecting the distribution of residual stress in this material [11]. Using the ENGIN-X strain diffractometer at the Science and Technology Facilities Council's (STFC) laboratory (Swindon, UK), Faisal et al. clearly demonstrated the residual stress distribution in Mo-C-based anode layers [12]. Based on their research on the diffractometer STRESS-SPEC at Forschungsneutronenquelle Heinz Maier-Leibnitz (FRM II, Garching, Germany), Zaeh et al. investigated the residual stress in selective laser melting, combining with

the finite element simulation method [13]. Xu et al. developed a procedure for residual stress evaluation with high stereographic resolution on the diffractometer TAKUMI at the Japan Proton Accelerator Research Complex (J-PARC, Ibaraki Prefecture, Japan). The texture and stress of a high-strength martensite-austenite multilayered steel were measured to examine the reliability of this procedure [14]. Li et al. measured the pole figure of warm-rolled Zircaloy-4 plate using a Neutron Texture Diffractometer (NTD) in the China Advanced Research Reactor (CARR) at the China Institute of Atomic Energy (CIAE, Beijing, China), and the result shown high reliability [15]. Moreover, there are other neutron scattering instruments that continue to contribute to the study of residual stress, such as the VULCAN at Oak Ridge National Laboratory (ORNL, TN, USA), the Fourier Stress Diffractometer (FSD) at the Joint Institute for Nuclear Research (JINR, Dubna, Russia), the DIANE at Laboratory Leon Brillouin (LLB, Saclay, France), and the Residual Stress Instrument (RSI) at the Korea Atomic Energy Research Institute (KAERI, Daejeon, South Korea).

This article specifically presents the recent progress of the Residual Stress Neutron Diffractometer (RSND) at the Key Laboratory for Neutron Physics of the Chinese Academy of Engineering Physics, Institute of Nuclear Physics and Chemistry, Mianyang, China. Neutron diffraction can be carried out under various experimental conditions with multiple pieces of equipment, such as a loading frame, manipulator, and coupling system integrated onto the RSND. For instance, the RSND can be used to effectively measure the texture and in situ evolution of the lattice structure of samples, which helps to reveal the deformation mechanism and thermodynamic properties of materials and components. Various materials with crystal structure, such as steels, aluminum alloys, titanium alloys, zirconium alloys, and superalloys can be studied with the RSND. The structure and internal stress in components such as turbine discs, blades, and welding parts are also able to be studied by neutron diffraction, offering valuable results of multiple reflections of poly and single crystal materials. In general, three types of experiments can be performed: (I) residual stress measurement, (II) in situ observation, and (III) texture analysis.

Remarkably, some complicated experiments, such as high-temperature in situ tension tests, texture measurements, and single crystal measurements, have been successfully performed. Among them, analysis for single crystal materials and components was considered a technical obstacle. It is now achievable, using our in situ neutron diffraction method, which provides new information for the study of structural evolution and performance changes with respect to single crystal materials and components.

## 2. Technical Characteristics of the Residual Stress Neutron Diffractometer at the China Mianyang Research Reactor

The RSND is located in the reactor hall of the China Mianyang Research Reactor (CMRR) in Mianyang, Sichuan Province. The fluxes of thermal and cold neutrons of the reactor are $2.4 \times 10^{14}$ n/(cm$^2$s) and $1 \times 10^9$ n/(cm$^2$s), respectively. The RSND, as shown in Figure 3, is one of eight neuron scattering instruments in the CMRR [6]. Its main parts, including a monochromator, a sample stage, and a detector, are marked out in Figure 1. Table 1 lists some of its basic parameters. The residual stress neutron diffraction measurement is a method which can nondestructively measure the internal stress and inner structure of materials and components.

The maximum neutron flux at the sample position is about $4.7 \times 10^6$ ncm$^{-2}$s$^{-1}$ (at the neutron wavelength $\lambda$ = 1.58 Å with a reactor power of 20 MW), which meets the experimental requirements for most measurements of crystal materials. A double focusing monochromator with a reflection of (311) for a silicon single crystal is employed. With the change of the take-off angle (~50–120°), the neutron wavelengths vary from 0.89 Å to 2.82 Å, which is adjustable for experiments requiring different gauge volumes. To improve the quality of the incident and diffracted neutron beams, the oscillating radial collimators are used with a transmissivity of ~90% for neutron beams and the Gaussian rocking curve.

The sample stage has five degrees of freedom, which are *X*, *Y*, *Z*, rotation, and tilt. The variable range is ±0.3 m for the *X*- and *Y*-axes and 0.5 m for the *Z*-axis, with the precision of 0.1 mm, while the

rotation angle ranges from 0° to 360° with the precision of 10′. The maximum loading capacity of the sample stage is about 500 kg. Due to the high loading capacity, many instruments such as a loading frame and rocking goniometer can be integrated into the sample stage. Therefore, complex neutron experiments such as texture measurements, single crystal experiments, and in situ tests are possible.

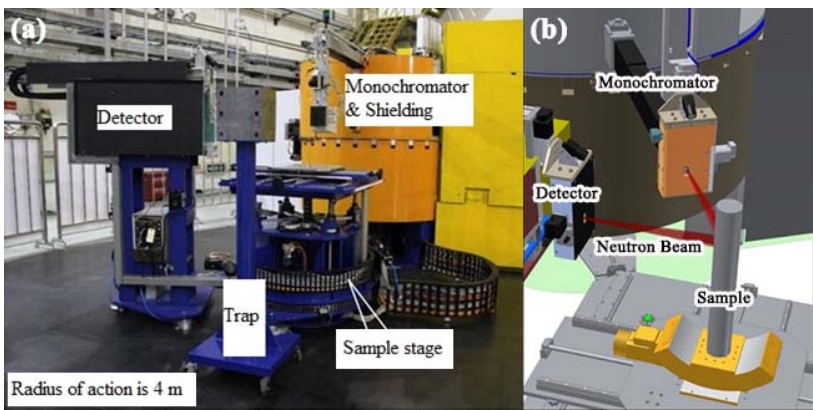

**Figure 3.** (**a**) The layout of the Residual Stress Neutron Diffractometer (RSND) with the main parts marked; (**b**) schematic diagram of neutron diffraction in the RSND.

**Table 1.** Comparison of the basic parameters between the RSND at the China Mianyang Research Reactor (CMRR) and the Residual Stress Diffractometer (RSD) at the China Advanced Research Reactor (CARR).

| Parameters | RSND at CMRR | RSD at CARR [16] |
|---|---|---|
| Max. neutron flux at sample position | $4.7 \times 10^6$ n/(cm$^2$s) | $2.7 \times 10^7$ n/(cm$^2$s) |
| Neutron wavelengths | 0.89–2.82 Å | 0.9–2.7 Å |
| Monochromator take-off angle | 50–120° | 40–110° |
| Optimized resolution $\Delta d/d$ | 0.18% | ~0.2% |
| Diffraction angle range | 0–145° | 3–120° |
| Calibrated efficiency | 72.1% | - |
| Max. loading capacity | 500 kg | 200 kg |
| *X*-axis and *Y*-axis adjustable distance of sample center | ±0.3 m | ±0.2 m |
| *Z*-axis adjustable distance of sample center | 0.5 m | - |

The He$^3$ two-dimensional (2D) position sensitive detector is employed to measure the diffracted neutron beam with a calibrated efficiency of 72.1% (at $\lambda$ = 2.31 Å). The effective detection area is $200 \times 200$ mm$^2$ and the spatial resolution is about $1.8 \times 1.8$ mm$^2$. The resolution of the diffractometer $\Delta d/d$ is measured to be 0.18% (at $\lambda$ = 2.31 Å) and the accuracy of the strain $\Delta\varepsilon$ is about $\pm 5 \times 10^{-5}$. The variable range of the diffraction angle is from 0° to 145°, satisfying the experimental requirements of various lattice reflections for most crystal structures. By changing the incident neutron wavelength, the measured gauge volume can be defined to be cubic, which increases the reliability of the experimental results. Additionally, to accommodate different kinds of samples, the distance between the sample and detector is adjustable. This makes the detector receive a strong beam of the scattered neutrons.

Compared with the Residual Stress Diffractometer (RSD) at the CARR in Beijing (Table 1), the technical characteristics of the RSND at the CMRR show great superiority, including a more flexible neutron wavelength, wider diffraction angle range, and greater loading capacity.

## 3. Environmental Instruments Attached to the Residual Stress Neutron Diffractometer

Neutron diffraction is a crucial test method in many fields concerning engineering, and residual stress neutron diffraction measurement is a unique method for reliability assessment through the

measurement of stress distribution inside materials and components. With the increasing complexity of test requirements, the experimental techniques have been improved. We have developed a number of components coupled to the instrument system. In addition, some original components, such as a shielding system, have also been optimized.

As shown in Figure 4, a loading frame is integrated for in situ uniaxial tensile and compressive neutron diffraction. The external force can load up to 15 kN, while the temperature can rise up to 500 °C, heated by a furnace, for in situ experiments. The tensile and compression rates can be controlled from 0.2 to 20 mm/min, while the heating rate is about 30 °C/min with the precision of $\pm 3$ °C and an effective heating range of 25 mm. The samples are sectioned into a specimen fixed into the stress rig. The measured crystal reflections are parallel to the loading direction with a strain extensometer recording the instantaneous strain of the specimen. To reduce the offset of the measured point, the unidirectional tensile loading function is applied while the crossheads move, with same displacement towards two sides. With the help of this instrument, the stress- and/or temperature-induced transformation can be measured in situ.

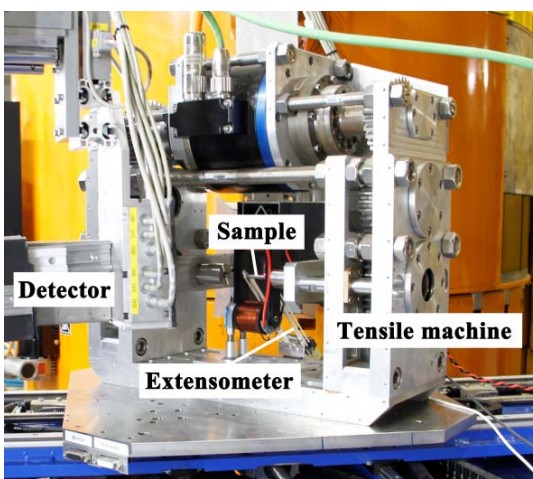

**Figure 4.** The front view of the loading frame integrated on the RSND.

To realize the measurement of radiated samples and the texture of small samples, a manipulator is available, as shown in Figure 5a. The manipulator is able to automatically catch and rotate the sample when it is connected to a remote-controlled program on the computer. Additionally, if the measured material contains elements that are easily activated by neutrons, such as Mn, Ni, and Au, the manipulator helps to relieve manual operation, ensuring the safety of experiments and preventing harm to operators' health. Though the texture of crystal samples can be roughly measured when the manipulator is incorporated, the test precision is not satisfactory, especially with large samples. Therefore, the Kappa goniometer with more accurate sample handling capabilities was developed. As shown in Figure 5b, this instrument is designed especially for the texture measurement of large-scale materials and components. The schematic diagram of its geometry is illustrated in Figure 2. The sample is fixed on the base. As a rotational relationship is shown in the diagram, $\varphi$ and $\chi$ respectively stand for the sample's rotation and tilt, whereas $\Omega$ represents the sample stage's rotation. The position of the diffraction volume gauge is adjusted to overlap with the center of the circle. Therefore, the measured region does not change during sample rotations. By changing the $\varphi$ and $\chi$ at suitable intervals, the texture of the sample can be measured. For some special materials, such as single crystal materials, the specific crystal planes can also be found by rotating and tilting the sample with neutron diffraction. In other words, the RSND has the ability to measure the lattice structure of single crystal samples, which is a challenge for other measurement methods [17].

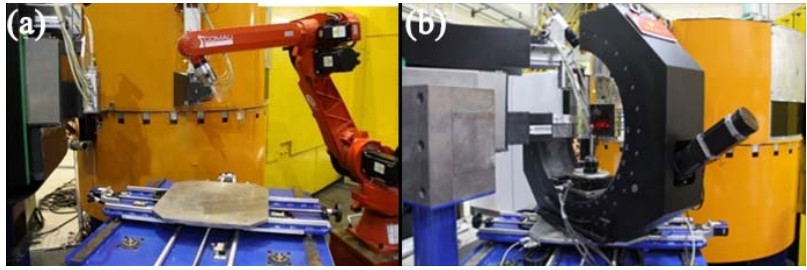

**Figure 5.** (**a**) The manipulator on the RSND; (**b**) the Kappa goniometer on the RSND.

Combining the advantages of the in situ loading frame and the texture measurement instrument, a new in situ thermomechanical and texture coupling system was developed (Figure 6a). As illustrated in Figure 6b, $\varphi$ and $\chi$ respectively represent the sample's rotation and tilt, while $\Omega$ stands for the sample stage's rotation. The texture of samples can be analyzed by rotating and tilting them during test procedures. A tensile machine was integrated into the system for in situ tensile experiments, with a maximum loading of 50 kN. To minimize the obstruction of the neutron beam while rotating the sample, a halogen spot heater was assembled for an in situ high-temperature experiment with a maximum temperature of ~1000 °C. The focus of the light beam was also at the center of the circle, with a thermocouple recording the temperature. Therefore, only the diffraction volume gauge needs to endure high temperatures, while other parts of the instrument remain in at ambient temperature in order to ensure the long-term stability of the equipment.

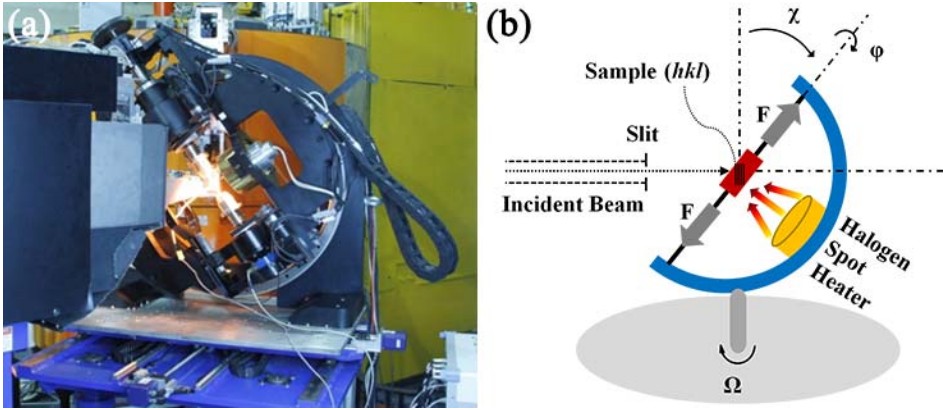

**Figure 6.** (**a**) The in situ thermomechanical and texture coupling system on the RSND; (**b**) the preliminary sketch of the geometry where the main parts are marked.

This coupling system is reliable in practical application. The in situ measurement mode can effectively track the structural evolution in test samples, overcoming problems of structural change during cooling and relaxing that exist with ex situ methods. Moreover, by rotating and tilting the sample, different directions of lattice planes can be measured. For example, when $\chi$ is 0, reflections parallel to the tensile stress are measured. Conversely, when $\chi$ is 90°, the evolution of planes perpendicular to the tensile stress is obtained. Especially for the measurement of single crystal materials, by rotating and tilting the sample, multiple crystal planes can be measured in situ with the high-temperature loading applied. Some planes sensitive to external stress such as {311}, {220} can also be measured in situ. Compared with other in situ neutron diffraction studies where the sample orientation of a single crystal material is fixed [18–20], this method is flexible and able to adjust the status of samples with multiple reflections measured at once. Following the structural evolution of tested samples in the in situ experiment, the diffraction signal may show slight deviations from Bragg's law, which may be beyond the scope of traditional methods of neutron diffraction. The RSDN, on the

other hand, is able to adjust the orientation and position of samples to ensure that the measured plane satisfies Bragg's law. Therefore, the evolution of structure in single crystal materials and components can be successfully tracked by in situ neutron diffraction.

Furthermore, in order to improve the experimental results, slits and shielding were further optimized. Al-Gd alloy was used as the material of the slits (Figure 7a). This alloy exhibits good shielding from neutrons and $\gamma$ rays, while it also shows good mechanical and thermal properties. The working temperature of the slits is about 500 °C, which ensures that they stay close to the sample surface even during in situ high-temperature experiments. The shielding for the detector is shown in Figure 7b. The shielding cavity is made of $B_4C$-Aluminium alloy, which further reduces the background. With the abovementioned optimizations, the experiment results from the neutron diffraction were considerably improved.

For further developments of the RSND, an advanced in situ experimental system with temperature-tension-torsion coupling is under consideration (Figure 8). This system is designed to meet the needs of some complex in situ experiments. The available experimental temperature ranges from −100 °C to 1000 °C. The maximum loading is 100 kN, while the maximum torsion is 50 Nm. When these parameters are realized, the RSDN will have the capacity to fulfill different requirements for most in situ mechanics experiments, providing more information and a deeper understanding of the performance of materials and components.

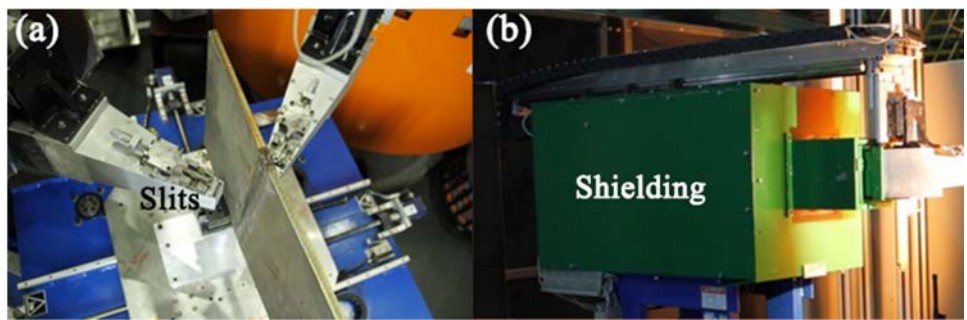

**Figure 7.** (**a**) The optimized slits for incident and diffraction neutron beams; (**b**) the optimized shielding for the neutron detector.

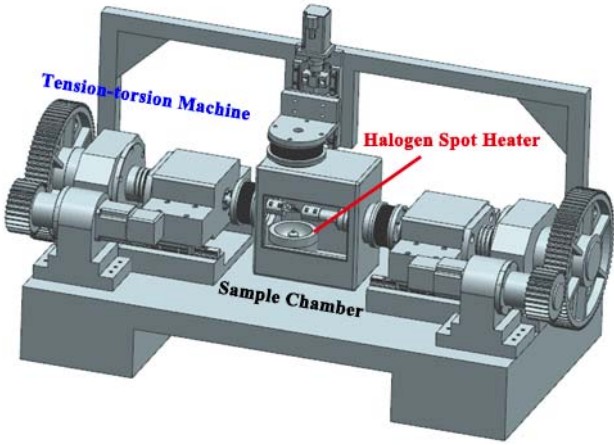

**Figure 8.** The layout of the in situ temperature-tension-torsion coupling instrument.

With the aforementioned development of the RSND, it is capable of performing some complicated experiments, such as multi-scale stress tests, high-temperature in situ tension tests, texture measurements, and single crystal measurements.

## 4. Application Research Examples Using Residual Stress Neutron Diffractometer

Residual Stress Neutron Diffractometer is a pragmatic tool for accessing the structure and distribution of the stress inside materials and components for reliability assessments. When equipped with other instruments, such as a loading frame, Kappa goniometer, and coupling system, the RSND can also be used to effectively measure the texture and in situ evolution of the lattice structure of samples.

Residual stress as a prevalent phenomenon in engineering materials and components, and has great impact on their properties. Therefore, the measurement of residual stress is of great significance in engineering. Wu and Sun et al. conducted many studies of the DD10 single crystal superalloy using neutron diffraction measurement [21,22]. Their investigation found that the evolution of the microstructure and the residual stress in DD10 superalloy showed correlation for different creep stages. For the primary stage of creep, the rate of dislocation multiplication was slow, while the residual stress in the superalloy increased. With the progress of creep, the dislocations largely multiplied to accommodate the deformation of the superalloy. Therefore, the residual stress gradually decreased [23]. The user from Shanghai Commercial Engine measured the 3D residual stress distribution of IN718 turbine discs, which provide guidance on the optimization of processing techniques of polycrystal superalloys. Han et al. conducted studies on the welding properties of nuclear reactor coolant pipe [24,25]. To meet the high-performance requirements, the base metals of the pipe were SA508 and 316L stainless steel, while Inconel 52 solid wire was chosen as the filler metal. The welding of dissimilar metals was the weak and complicated part of the pipe. Therefore, the properties of the welded part were of great significance. By neutron diffraction measurement, obvious stress concentration was observed inside the welding zone. Peng et al. investigated the distribution of residual stress in aircraft fuselage skin. Through neutron stress analysis, the residual stress profile measured along the depth of the alloy sheet was carried out. The results showed that the hoop stress in the center is small, whereas that near the surface was large. The measured neutron diffraction results showed good agreement with the simulation results by finite element method (FEM) and further modified the deformation theory of component machining [26].

To investigate the structural evolution of materials under specific circumstances, in situ neutron diffraction measurement is one of the advanced methods. Li et al. measured the deformation modes of Zr-4 alloy in situ. The internal stresses of the alloy and its deformation mechanisms were calculated by elastic-plastic self-consistent (EPSC) simulation [27]. Xu et al. observed the deformation behavior of nickel aluminum bronze (NAB) alloy in situ. The compressive internal stress in the $\alpha$ matrix and tensile internal stress in the $\kappa$ phase in the elasto-plastic region were measured. The results showed that with the increase of the deformation degree, the internal stress concentrates near the $\alpha/\kappa$ interface and gradually increases [28]. Zou et al. employed in situ neutron diffraction to investigate the deformation mode transition of Mg-3Li alloy. They found that the anelasticity from {10–12} twins induced the hysteresis loop of stress-strain response by uniaxial loading-unloading tension. The lattice strain in {10.0} was saturated at about 84.5 MPa, and then it abruptly decreased to ~132.6 MPa, which suggested that the state of prismatic slip was around the yield point. The load redistribution continuously occurred between soft and hard grain orientations [29].

By using a Kappa goniometer and manipulator, the texture of materials and components can be effectively measured. In the study of the radiation effect of Zr-4 alloy, we found that the intensity for both {00.2} and {10.0} pole figures grew stronger after radiation, which indicated that the texture of the alloy gradually changed. The results provided useful information to understand the mechanism of radiation-induced damage. Li et al. observed the evolution of the texture of Zr-4 alloy in situ during compression deformation. With the application of compression, the texture of Zr-4 alloy did not show obvious change [30]. Chen et al. investigated the evolution of the microstructure and texture for $AA6063/TiB_2$ composites using instruments on the RSND. The nanosized $TiB_2$ particles were redistributed in fine-grained $AA6063/TiB_2$ composite. The texture analysis showed that the nanosized $TiB_2$ particles in the composite blocked the movement of grain boundaries, retarding the process of dynamical recrystallization. Therefore, the deformation texture was preserved [31].

With continuous improvement and optimization, the RSND will be applied to more complex scientific research. Recently, some representative work has been achieved, using the RSND, by the group of the Institute of Nuclear Physics and Chemistry Key Laboratory of Neutron Physics.

### 4.1. Quenching-Induced Residual Stress in GH4169 Superalloy

During the manufacturing process of GH4169 nickel-based polycrystal superalloy, the quenching process is indispensable. However, due to the high temperature gradient, the residual stress is largely introduced in the superalloy. This kind of high residual stress undoubtedly has a significant effect on the performance of the superalloy [32]. Therefore, it is of great importance to measure the residual stress inside material accurately. In this study, the distribution of residual stress in GH4169 alloy after quenching was characterized by neutron diffraction [33].

The nominal chemical compositions (wt. %) of GH4169 superalloy were shown as following: 18.05Cr, 18Fe, 5.42Nb, 2.90Mo, 0.91Ti, 0.48Al, 0.10Si, minor C and Mn, and balance Ni. The cylindrical sample was heated for solution treatment at 980 °C for 2 h following quenching. After the heat treatment, only some δ phase precipitated in the austenitic matrix (Figure 9). The (311) reflection which was suitable to measure the long-range internal stress in materials, and thus was chosen as the diffraction crystal plane [21]. The neutron wavelength was 1.593 Å, and thus the diffraction angle was ~90° to define a cubic gauge volume of $2 \times 2 \times 2$ mm$^3$. The elastic modulus $E$ and Poisson's ration $v$ of (311) is 200 GPa and 0.3, respectively [34]. The size of this sample was 40 mm in diameter and 40 mm in height, and the representative measured points were chosen as shown in Figure 10a. Region A was 3 mm from surface and 3 mm from side, region B was 20 mm from surface and 3 mm from side, and region C was at the center of the sample. To obtain the value of $d_0$, a stress-free cubic sample was measured (Figure 10b).

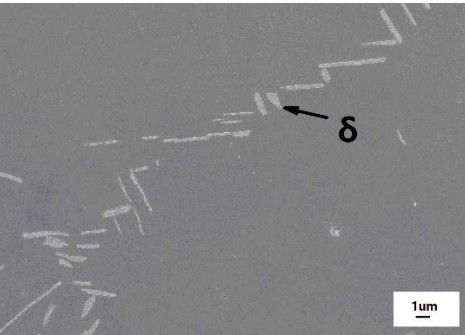

**Figure 9.** The microstructure of GH4169 superalloy after heat treatment where the δ phase is marked [33].

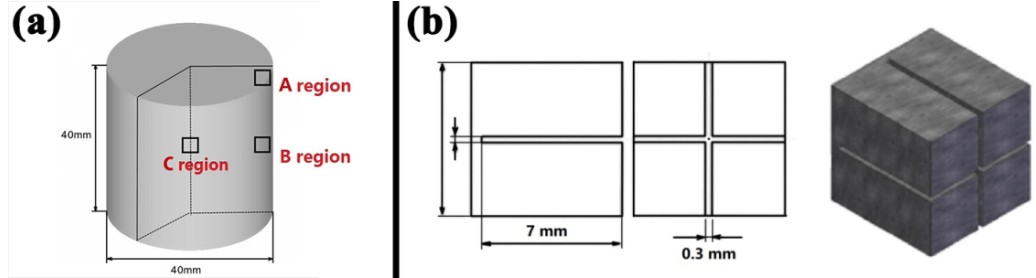

**Figure 10.** (**a**) The representative measured points in the GH4169 superalloy sample, where A region is 3 mm from surface and 3 mm from side, B region is 20 mm from surface and 3 mm from side, and C region is at the center of the sample; (**b**) the stress-free sample with a cross cut for the $d_0$ measurement [33].

The neutron diffraction data and profiles are shown in Figure 11. The hoop, axial, and radial directions of normal stress are measured. The raw data are fitted by a Gaussian function and the uncertainty of fitting the peak position is about 0.005°, which is then calculated as the error for strain and stress. Due to the existence of residual stress inside the sample, the *d*-spacing of the lattice will be compressed or stretched. Based on Bragg's law: $n\lambda = 2d \sin \theta$, the change of the *d*-spacing of the lattice will result in a diffraction peak position shift. Therefore, the diffraction peak position of the quenching sample will contain residual stress shifts to some extent, compared with the initial stress-free sample. The degree of the shift directly reflects the magnitude of residual stress inside the sample.

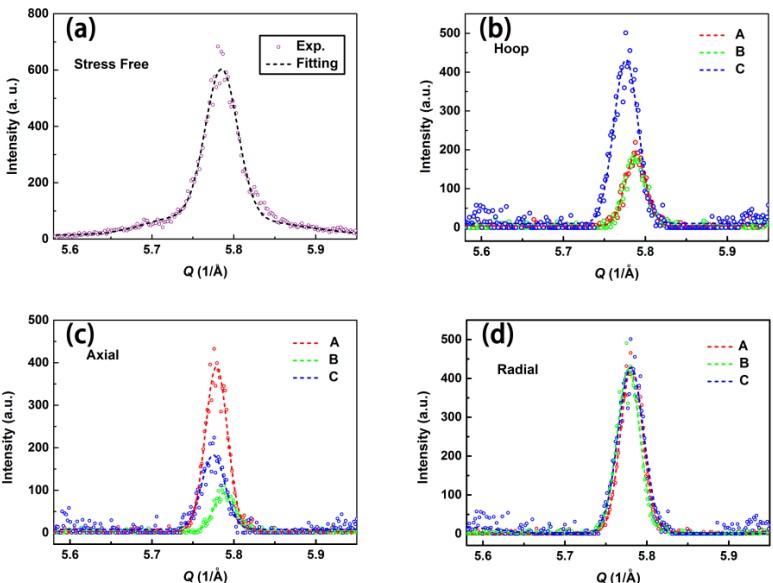

**Figure 11.** The neutron diffraction data and the profiles of (**a**) the stress-free sample, as well as the (**b**) hoop, (**c**) axial, (**d**) radial directions of the GH4169 superalloy sample, respectively. The hollow dots stand for raw data, while the dotted lines stand for fitting profiles. The red line stands for A region, the green line stands for B region and the blue line stands for C region [33].

Using Bragg's law, the lattice parameters of (311) can be determined. With the *d* for each direction and stress-free $d_0$ obtained, the residual stresses for different directions of regions can be calculated using Equation (1). According to the results shown in Table 2, at surface region A, the hoop compressive residual stress is large, while the residual stresses of the radial and axial directions are relatively small. However, at region B, the compressive stresses of the hoop and axial directions are close to the value of about −350 MPa, whereas the radial stress is small. At region C, residual stresses of all three directions are tensile stresses with the value of ~400 MPa. This phenomenon suggests that during the quenching process, the large thermal gradient produces nonuniform residual stress. The temperature of the center of the sample decreases with the volume contraction. However, the temperature of the surface is much lower than that of the center. Therefore, the deformation of the surface is less severe than that in the center, which confines the contraction effect of the center. The stress at the center of the sample gradually increases. With the increase of contraction in center, the stress state of the center of the sample becomes tensile, while that of the surface becomes compressive.

By researching the quenching-induced residual stress in GH4169 nickel-based polycrystal superalloy, some meaningful conclusions can be drawn. The results showed that after quenching, the residual stress in the center of the sample was a triaxial tensile stress state for about 400 MPa, whereas that at the surface of the sample was a uniaxial or biaxial stress state for about −300 to −400 MPa. This indicated that during the quenching method, a large thermal gradient was produced between the center and the surface of the sample, which led to huge thermal stress. Owing to the high thermal stress, nonuniform plastic deformation occurred. As a result, a self-balanced residual

stress was produced. This result agrees with FEM results in References [35,36] concerning steel and aluminum alloy.

**Table 2.** The diffraction data and residual stress in regions A, B, and C for the hoop, axial, and radial directions of the GH4169 superalloy sample. The errors are shown in parentheses [33].

| Region | 2θ (°) | | | Residual Stress (MPa) | | |
|--------|--------|--------|--------|--------|--------|--------|
|        | Hoop | Axial | Radial | Hoop | Axial | Radial |
| A | 94.495(7) | 94.296(6) | 94.260(5) | −243(23) | 4(16) | 49(18) |
| B | 94.491(7) | 94.509(9) | 94.180(6) | −357(23) | −380(21) | 30(21) |
| C | 94.243(8) | 94.243(8) | 94.213(16) | 398(30) | 435(29) | 398(30) |

### 4.2. Evolution of Mechanical Properties for Polycrystalline Beryllium

Due to its excellent interaction with plasma, the metal beryllium (Be) with a hexagonal close-packed (hcp) structure is widely used as the first wall material of a fusion reactor [37]. However, polycrystalline beryllium has some shortcomings of mechanical properties, such as low elongation, brittle fracture, and obvious asymmetry in mechanical properties between tension and compression, all of which limit its use. Therefore, much work has been done to research the mechanical properties and deformation mechanism of polycrystalline beryllium. This work investigates the mechanical properties of polycrystalline beryllium with various initial microstructures. By the self-coordination of microstructures, the mechanical properties are modulated [38].

The purity (wt. %) of the polycrystalline beryllium sample was >99%, with minor amounts of C and Fe. Three samples with different initial microstructures were pretreated as follows. Sample 1 was quasi-statically compressed to the strain of ~0.2 with the rate of $10^{-3}$/s at room temperature. Sample 2 was quasi-statically compressed to the strain of ~0.2 with the rate of $10^{-3}$/s at 600 °C. Sample 3 was dynamically compressed to the strain of ~0.2 with the rate of $10^3$/s at room temperature. The microstructures of these three samples are shown in Figure 12. After different processes of compression, the microstructure of sample 1 presented a small number of microvoids, sample 2 was relatively intact with few microvoids, and sample 3 showed many microvoids of a large size. This phenomenon indicates that under the high loading rate, the deformation of the sample cannot continue to adapt to the speed of loading, which produces stress concentration. Therefore, many microvoids were generated in sample 3.

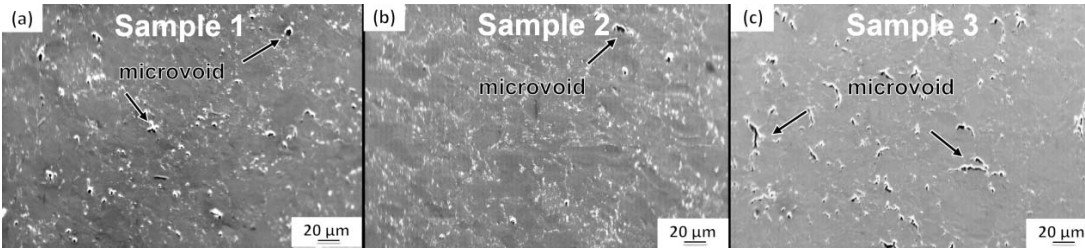

**Figure 12.** (**a**–**c**) The microstructure of polycrystalline beryllium after different pretreatments for samples 1, 2, and 3, respectively. The microvoids are marked in the images [38].

The in situ neutron compression experiments for polycrystalline beryllium samples with different initial microstructures were performed using the loading frame on the RSND. The neutron wavelength for measurement was 1.587 Å, defining a cuboid gauge volume of $2 \times 2 \times 2$ mm$^3$. The compression rate was set as 0.5 mm/min and the diffraction data of (00.2), (10.2), and (11.0) were continuously recorded. The value of $d_0$ was chosen from the unloaded state of each sample. The evolutions of the lattice strain calculated from Equation (1) for each sample are exhibited in Figure 13. For sample 1, the lattice strain of (00.2) is larger than the other two measured planes, which indicates that (00.2)

bears stress first (Figure 13a). However, the planes of sample 2 show a similar stress response, and with compression increase, (11.0) gradually shows the larger lattice strain (Figure 13b). The situation of response for planes in sample 3 is similar to that in sample 1 (Figure 13c). The changes of the lattice strain of (11.0) and (10.2) are small, while that of (00.2) is relatively large. This phenomenon demonstrates that (00.2) in sample 3 is principally subjected to external stress.

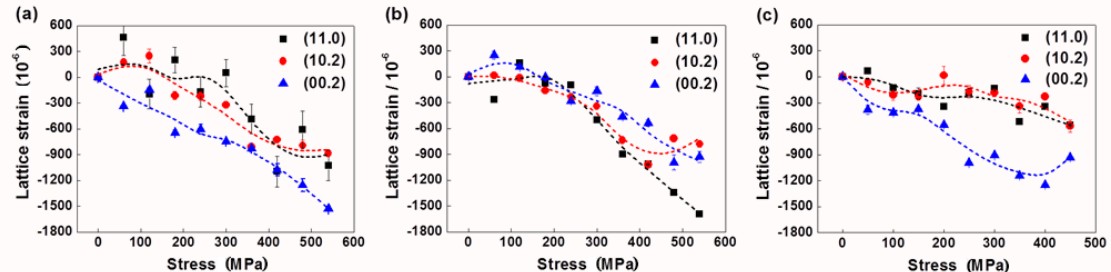

**Figure 13.** (a–c) The response of lattice strain during the in situ loading for different planes of polycrystalline beryllium samples 1, 2, and 3, respectively. The discrete symbols stand for raw data, while the dotted lines stand for fitting curves [38].

Combining all results of the microstructure and evolution of the lattice strain for polycrystalline beryllium samples with various initial states, some features of the evolution of mechanical properties can be revealed. The sample pretreated quasi-statically by room temperature compression shows that (00.2) is primarily subjected to pressure. Due to the nonuniform distribution of stress, it shows a relatively visible deformation hardening effect. The sample pretreated dynamically by room temperature compression also has (00.2) bearing stress first. However, the microvoids shared the stress and delayed the deformation hardening. The planes of the sample that was pretreated quasi-statically by high-temperature compression are equally stressed. With the stress increasing, the stress on (11.0) gradually increases. This phenomenon suggests that the grains in this sample are of random orientations during the high-temperature heating process. To reveal the orientation distribution of grains in different samples, the orientation distribution function (ODF) for samples 1–3 was further measured (Figure 14), and corresponded to the aforementioned results. Obviously, compared with sample 1, sample 2 has a more random distribution. However, sample 3 prefers special orientations of strong (00.2) fiber texture.

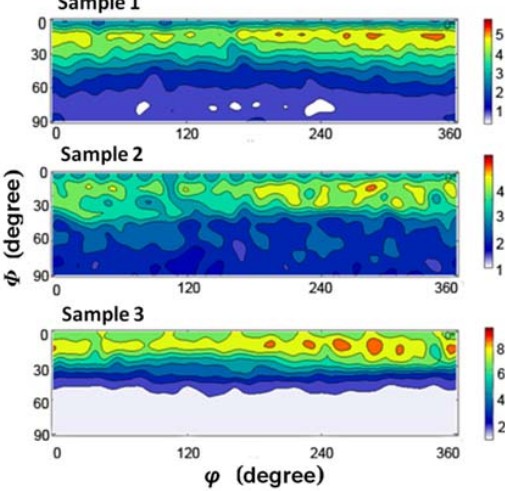

**Figure 14.** The orientation distribution function (ODF) of polycrystalline beryllium samples 1–3 after different pre-deformations at $\theta = 0°$ [38].

### 4.3. In Situ Observation of DD10 Single Crystal Superalloy

The single crystal superalloy shows excellent high-temperature and high-stress mechanical properties, and is widely used as the turbine blade material in aircraft engines [39]. During the heat treatment, the $L1_2$ cubic $\gamma'$ phase separates out from the face-centered cubic (fcc) $\gamma$ phase matrix regularly. To reveal the correlation between the proprieties and evolution of the structure for a single crystal superalloy in service, in situ diffraction measurement is necessary. However, owing to the limitations of the test equipment, the changes of lattice planes in single crystal materials is difficult to track. Moreover, little research has focused on the in situ measurement of structural evolution under high-temperature and high-stress conditions [40,41]. In previous in situ neutron diffraction studies, individual few planes, such as (002), (020), and (200), were analyzed [18,42]. The external stress-sensitive planes, such as {311}, {220}, were not considered. To understand the evolution of the lattice structure in single crystal superalloys adequately under high-temperature and high-stress conditions, a new method to observe the lattice change for multiple lattice planes in situ was gradually developed on the RSND. In this work, the lattice parameters during elastic tension were studied by in situ neutron diffraction. Multiple planes, such as {002}, {220}, and {311}, were measured. Meanwhile, the change of stress inside the sample was calculated, which corresponded with the evolution of lattice distortions.

The single crystal nickel-based superalloy DD10 was used in this research. Its nominal chemical compositions (wt. %) were shown to be the following: 13.0Cr, 5.3Al, 4.1 Ti, 4.0Co, 3.9W, 3.1Ta, 2.3Mo, with minor B and C, and balance Ni. The sample was cut along the [001] orientation, and it was machined into a tensile specimen. The following standard heat treatment was applied: 1250 °C/ 3 h/air cooling (AC) + 1100 °C/5 h/AC + 870 °C/24 h/AC. Then the sample was fixed on the coupling system as shown in Figure 15a. The sample was heated up to 500 °C within 1 h, holding for 2 h for thermal stress relaxation. Holding the temperature at 500 °C, external stress $\sigma$ was applied gradually up to 760 MPa within 1 h by the tensile machine. The condition of high temperature and stress lasted for about 10 h for the in situ neutron diffraction measurement, followed by unloading and air cooling (Figure 15b). The neutron wavelength $\lambda$ was 1.593 Å. By adjusting the $\varphi/\chi$ rotation relationship and the $2\theta$ diffraction angles, the diffraction data of {002}, {220}, and {311} were successively measured in situ. The measured diffraction reflections, which included the results of $\gamma$ and $\gamma'$ phases, were fundamental reflections. Therefore, the diffraction profile should be deconvolved reasonably. The principles of deconvolution were discussed in Reference [22] in detail.

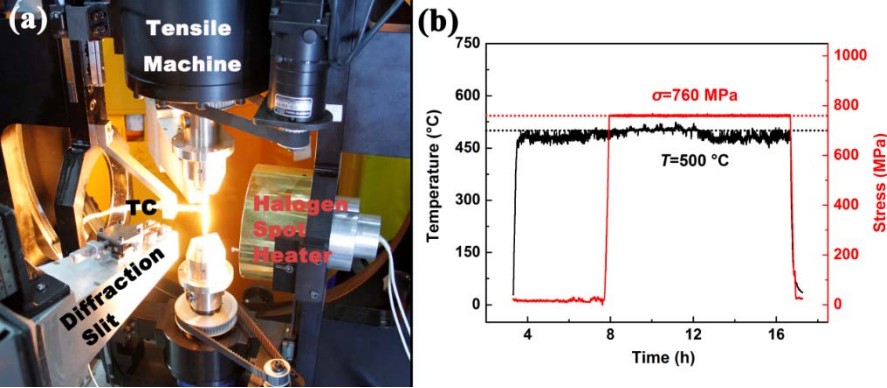

**Figure 15.** (**a**) The picture for the test of the in situ thermomechanical and texture coupling system, where all main parts are marked; (**b**) the heating and tensile curves for the DD10 single crystal superalloy sample [43].

The evolutions of lattice parameters for multiple planes are shown in Figure 16. After the sample was heated to 500 °C, the lattice parameters increased due to the thermal expansion. The close values

of parameters indicate that the thermal stress is almost relaxed. Meanwhile, it hints that the residual stress in the sample might also relax slightly. The tensile response for the planes of phases shows anisotropy. The planes, such as (002), (022)/(202), and (113), which are vertical to the [001] tensile direction to some extent show an increase of parameters as the loading is applied. Meanwhile other reflections, which are to some extent parallel to [001], exhibit a decrease of lattice parameters. Because the γ′ phase is stiffer than the γ phase at 500 °C [32], the parameters of the γ phase show greater change compared with that of the γ′ phase. As the time elapses, the parameters remain almost unchanged. This suggests that during the process of the experiment, only elastic strain occurs.

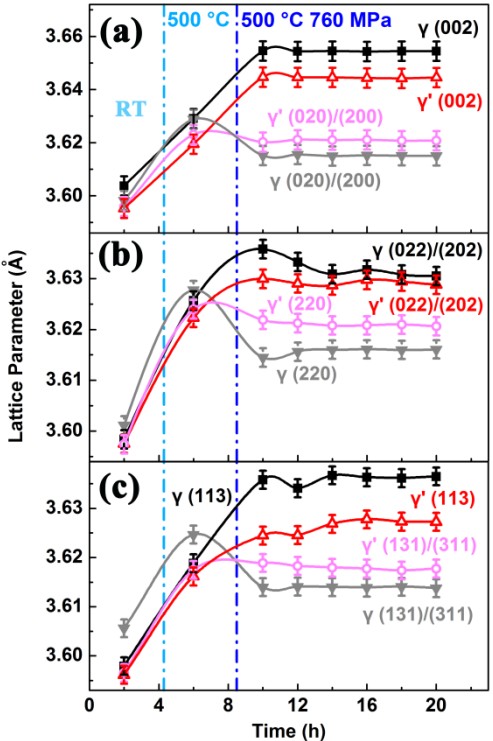

**Figure 16.** The evolution of lattice parameters for the (**a**) {002}, (**b**) {220}, and (**c**) {311} planes of DD10 superalloy. The solid symbols stand for the γ phase, while the hollow symbols stand for the γ′ phase [43]. RT: room temperature.

The evolution of macrostress is calculated using Equation (2). The stress tensor components can be obtained using Hook's law: $\sigma_{ij} = C_{ijkl}\varepsilon_{kl} = \frac{E_{hkl}}{1+v_{hkl}}\left(\varepsilon_{ij} + \frac{v_{hkl}}{1-2v_{hkl}}\varepsilon_{kk}\delta_{ij}\right)$, where $E_{hkl}$ is the Young's modulus of (*hkl*), $v_{hkl}$ is the Poisson's ratio of (*hkl*), and $\delta_{ij}$ is the Kronecker delta-function. In this alloy, the diffraction data of {311} reflections are used and the elastic coefficients $C_{11}$, $C_{12}$, $C_{44}$ for these planes are 162 GPa, 109 GPa, and 76 GPa, respectively [44]. Hence, the von Mises stress $\sigma_{Mises}$ is calculated, which is a commonly used engineering parameter to evaluate the equivalent stress in samples. As shown in Figure 17, the von Mises stresses are ~196 MPa in the sample at room temperature (RT). After holding at 500 °C for some time, the von Mises stresses relax to about 172 MPa. After the tensile stress is applied, the lattice stress increases to about 750 MPa, with few changes occurring after this value is reached. This agrees well with the elastic condition of the sample.

The evolution of the lattice parameters for multiple planes and the inside stress of DD10 superalloy were reasonably analyzed by in situ neutron diffraction under high-temperature and high-stress conditions. The diffraction results of multiple planes were measured by the in situ diffraction method for the first time. The results give a meaningful understanding of the structural evolution of single crystal materials, and some conclusions can be drawn. The tensile response for the planes of the γ and γ′ phases shows obvious anisotropy. As the γ phase appears to be less stiff compared with the

γ′ phase at 500 °C, the lattice parameters of the γ phase exhibit greater change as the tensile stress is applied. After the sample is heated to 500 °C, the values of the lattice parameters become closer to each other. This phenomenon results from the relaxation of the lattice stress inside the sample. The von Mises macrostress reduces from ~196 to 172 MPa. After the tensile stress is applied, the von Mises stress of the sample increases to about 750 MPa and remains almost unchanged as the tensile time elapses. Thus, the results of the evolution of lattice parameters and macrostress in the superalloy show consistency.

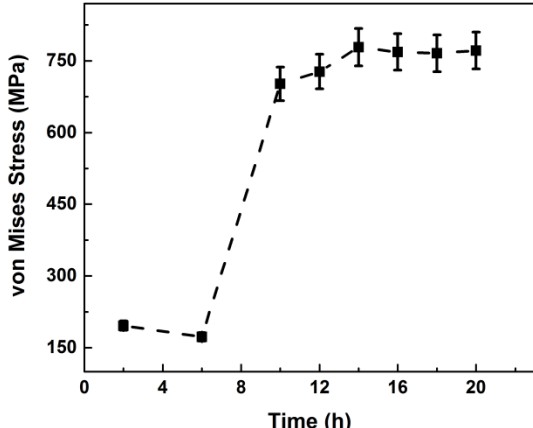

**Figure 17.** The evolution of the von Mises stress inside the DD10 single crystal superalloy [43].

## 5. Summary and Future Prospects

In conclusion, the RSND is a neutron diffraction experiment platform for the research of residual stress and material structure. Based on the introduction of different types of equipment, such as a loading frame, manipulator, Kappa goniometer, and in situ thermomechanical coupling system, the RSND can provide a great deal of structural information of materials and components. First, the internal residual stress inside the materials and components can be effectively measured, which is important for reliability assessments in engineering. Meanwhile, the texture and its evolution in materials can be successfully observed. Moreover, in situ neutron diffraction can provide meaningful data of materials and components under specific temperature and tension conditions. The structural evolution of samples can be tracked simultaneously. In particular, due to the breakthrough in the in situ measurement of single crystal materials, multiple families of crystal planes can be measured at once by the thermomechanical coupling system. The difficulties of single crystal neutron diffraction measurement are effectively solved. By the use of this coupling system, a deeper understanding of the deformation mechanism for single crystal components under specific circumstances can be achieved, providing guidance for the manufacture and processing of single crystal materials and components.

**Author Contributions:** Conceptualization: Y.C., H.L., B.P., J.G., B.C. and S.P.; Methodology: J.L., H.W., H.L., Z.Y., B.P., J.G., B.C. and S.P.; Formal Analysis: F.M., H.W., Z.L. and Z.Y.; Investigation: J.L., C.Z., Z.L., Z.Y., Z.Y., B.P., Y.H. and Y.T.; Data Curation: F.M., C.Z., H.W., Z.L., Z.Y., Z.Y., Y.H. and Y.T.; Writing-Original Draft Preparation: F.M.; Writing-Review & Editing: G.S. and Y.C.; Supervision: G.S., J.G., B.C. and S.P.; Project Administration: G.S., C.Z., B.P. and S.P.

**Funding:** These studies are supported by the National Key Research and Development Program of China (Grant No. 2017YFA0403703), the National Major Scientific Research Equipment of China (Grant No. 51727801), the research of in situ mechanical-thermal coupling for neutron scattering (2017BB02), the Science Challenge Project (Grant No. TZ2016004), and the Key Laboratory Foundation for Neutron Physics of CAEP (Grant No. 2015BB05).

**Acknowledgments:** These studies are supported by the National Key Research and Development Program of China (Grant No. 2017YFA0403703), the National Major Scientific Research Equipment of China (Grant No. 51727801), the research of in situ mechanical-thermal coupling for neutron scattering (2017BB02), the Science Challenge Project (Grant No. TZ2016004), and the Key Laboratory Foundation for Neutron Physics of CAEP (Grant No. 2015BB05).

**Conflicts of Interest:** The authors declare no conflict of interest.

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
