# Peer review of "Recent Progress of Residual Stress Distribution and Structural Evolution in Materials and Components by Neutron Diffraction Measurement at RSND"

_qubs, doi:10.3390/qubs2030015_

Round 1
Reviewer 1 Report
Please see attached file.

Author Response
Thanks for your letter and for the comments concerning our manuscript.
The response to the editors’ and the reviewers’ comments and main corrections in the paper are shown in the attachment.

Reviewer 2 Report
This paper presents the recent development of the residual stress measurement in Key Laboratory for Neutron Physics of Chinese Academy of Engineering Physics, Institute of Nuclear Physics and Chemistry, Mianyang, China. Multiple accessories such as Goniometer, in situ load frame, are integrated to the neutron diffractometer over the time to facilitate the stress measurement. It was concluded that after the integration of the thermomechanical coupling system and the breakthrough in the in-situ measurement the difficulties of single crystal neutron diffraction measurement are well overcome. The paper reads well and the flowline of the story is smooth. However, before accepting the article following needs to be addressed:
1. In page 2, line 63: Type I and type II residual stress appeared suddenly. From the reviewer perspective, it would be good if you introduce the type I and type II stress earlier with a schematic.
2. It is hard to visualize the symbols used in equation 2. Is it possible to introduce a schematic? Another suggestion would move the figure 4 here to visualize all the symbols.
3. In Table 1: Please change "Max. neutron flux" to "Max. neutron flux at the position of sample" to avoid confusion.
4. Please double check all the symbols for temperature and angles. I found there are some discrepancies.
5. The quality of figure 2 is poor. Please provide a better quality figure and also please mark the main elements in the figure.
6. Please mark the main features in figure 7.
7. Page 10 line 318: Please include the size of the gauge volume.
8. In figure 9: Please include the sample name "GH4169 superalloy" in the caption.
9. Please include a paragraph on page 11 explaining the shift in the peak position observed in figure 10. Why the peaks were observed at different values of Q for the different region of A, B, and C.
10. Please include the sample name "GH4169 superalloy" in the caption for figure 10 and table 2.
11. Please use different symbols rather than A, B, and C in figure 11 as you already have used these symbols to define different regions for the GH4169 superalloy. Also, the size of the gauge volume is missing for Polycrystalline Beryllium samples.
12. Please include the sample name "Polycrystalline Beryllium" in figure 13. Please include the sample name "DD10" in the figure caption of 14.
Overall the paper is well-written and will attract citations.
Author Response

(The authors gave the same response as above.)

Round 2
Reviewer 1 Report
Please see attached document.

Author Response
Thanks sincerely for your careful comments.
We submit a point-to-point response in the attached document.

Reviewer 2 Report
The authors made significant effort to address the reviewers comments. The paper has improved. However, with respect to the editor comments regarding residual stress measurement at KOWARI: you have included all the stress measurement in welded joint. Have you ever tried to measure residual stress in rivet? Stress measurement in riveted joint at Kowari has been showcased in ANSTO website as below: http://www.ansto.gov.au/ResearchHub/OurInfrastructure/ACNS/Industry/Services/Advancedmaterials/Self-PiercingRiveting/index.htm
It might be a good idea to include some discussion on this.
Author Response
Thanks for the comment. It is actually an interesting and meaningful idea to measure the residual stress in rivet. Therefore, we have discussed the results from KOWARI in the introduction part (Page 3, Line 97-101).
The full response letter is in the attached document.
